# Exploring the Mechanisms of Iron Overload-Induced Liver Injury in Rats Based on Transcriptomics and Proteomics

**DOI:** 10.3390/biology14010081

**Published:** 2025-01-16

**Authors:** Yujia Shu, Xuanfu Wu, Dongxu Zhang, Shuxia Jiang, Wenqiang Ma

**Affiliations:** 1Key Laboratory of Animal Physiology and Biochemistry, Ministry of Agriculture and Rural Affairs, College of Veterinary Medicine, Nanjing Agricultural University, Nanjing 210095, China; 2021107015@stu.njau.edu.cn (Y.S.); 2021807150@stu.njau.edu.cn (X.W.); 2022107030@stu.njau.edu.cn (D.Z.); jiangshuxia@sues.edu.cn (S.J.); 2MOE Joint International Research Laboratory of Animal Health & Food Safety, Nanjing Agricultural University, Nanjing 210095, China; 3Shanghai Frontiers Science Research Center for Druggability of Cardiovascular Noncoding RNA, Institute for Frontier Medical Technology, School of Chemistry and Chemical Engineering, Shanghai University of Engineering Science, Shanghai 201620, China

**Keywords:** proteomics, transcriptomics, iron overload, liver injury

## Abstract

Iron is both a necessary nutrient and a potential toxin for the body. The liver, as the central organ for maintaining overall iron homeostasis, is also the main organ responsible for oxidative stress caused by iron overload toxicity. However, the mechanism by which iron overload causes cellular damage is not yet fully understood. Transcriptomics and proteomics combined analysis can observe the correlation between RNA and proteins, providing a comprehensive understanding of biological questions, thoroughly exploring the pathogenesis of organisms, and precisely studying the expression patterns and regulatory mechanisms of important genes. Therefore, this study focuses on male SD rats and BRL-3A cells as research subjects. By combining transcriptomics and proteomics analysis, we investigate the mechanisms underlying iron overload-induced liver damage in rats.

## 1. Introduction

Iron is a vital trace element necessary for nearly all living organisms. Due to the fact that free iron ions readily accept or lose electrons, they can serve as essential cofactors for many enzymes and other active molecules. Iron has a dual function as it can both maintain various metabolic and physiological processes as well as trigger cellular damage induced by harmful molecules [1]. Hemochromatosis is a genetic condition characterized by an abnormal accumulation of iron within the body [2]. Iron builds up in parenchymal tissues due to excessive dietary absorption and disrupted recycling processes [2]. Excessive cellular iron induces reactive oxygen species, potentially resulting in organ damage [3]. Hemochromatosis is a disorder characterized by the absorption of excess iron, which can result in the manifestation of iron-induced toxicity. This toxicity affects vital organs, including the brain, heart, liver, pancreas, and bones [2]. Within the liver, alcohol consumption, viral hepatitis, and metabolic syndrome can exacerbate iron’s effects, leading to progressive liver disease. This occurs through the synergistic activation of hepatic stellate cells, which in turn promotes fibrogenesis [4]. Hepatic fibrogenesis results from elevated ROS production due to hepatic iron overload [5]. Enzyme polymorphisms related to ROS generation or degradation, like heme oxygenase (HO)-1, are crucial in influencing fibrogenesis due to iron overload [6].

HO-1, otherwise designated as heat shock protein 32, has been identified as the rate-limiting enzyme in the heme degradation pathway. This enzyme is responsible for the breakdown of hemoglobin into biliverdin, free iron, and carbon monoxide (CO) [7]. A plethora of evidence highlights the dual function of HO-1; the degradation of heme produces excess Fe^2+^, which accumulates free radicals through the Fenton reaction, resulting in lipid peroxidation [8]. The presence of excess iron in the free state within the body has been indicated as a factor that can induce cellular iron death [9]. However, an alternative perspective is also to be considered. HO-1 is regarded as a major antioxidant. A variety of studies have indicated that the activation of the Nrf2/HO-1 pathway and elevated HO-1 expression can elicit anti-inflammatory effects and impede inflammation [10,11]. The protective effect of HO-1 mostly relies on its metabolites, such as biliverdin and CO. Biliverdin is reduced to bilirubin by biliverdin reductase [12]. Biliverdin and bilirubin are bile pigments that possess antioxidant properties [12]. Endogenous CO functions as a second messenger, modulating a variety of physiological and pathological processes, including cell proliferation, inflammation, apoptosis, and tissue injury [13]. Furthermore, it has been documented that HO-1 upregulates ferritin (the iron-binding proteins) and FPN (iron transport protein), thereby reducing intracellular iron content and oxidative stress [14,15]. Nonetheless, there is limited understanding of its potential mechanisms in a liver iron overload model.

Long non-coding ribonucleic acids (LncRNAs) represent a category of RNA molecules that do not undergo translation into peptides and are distinguished by a sequence length that exceeds 200 nucleotides. LncRNA is crucial in various cellular processes, including dosage compensation, epigenetic and cell cycle regulation, and cell differentiation [16]. LncRNAs can play various roles, such as regulating transcription or affecting protein/RNA stability by binding to DNA, RNA, or proteins [17,18,19,20]. Research has demonstrated that in the context of rheumatoid arthritis, the overexpression of LINC00638 activates the Nrf2/HO-1 pathway, thereby suppressing inflammation and oxidative stress [21].

While LncRNA has been shown to activate HO-1 to mitigate oxidative stress, the precise function of HO-1 in liver iron overload models and its regulation by certain LncRNAs remain unclear. This study aims to conduct an in-depth study of HO-1’s involvement in iron overload-induced liver injury and thus identify its potential therapeutic approach to alleviate liver fibrosis in hemochromatosis. HO-1 has been identified as a candidate protein through TMT quantitative proteomics and protein–protein interaction (PPI) network analysis. Through the combined analysis of proteomics and transcriptomics, we identified Lnc286.2 as a potential transcriptional regulator of HO-1. We demonstrated that HO-1 and Lnc286.2 are upregulated in BRL-3A cells after FAC treatment, acting as regulatory factors for iron overload by modulating intracellular oxidative stress-related markers. By inhibiting Lnc286.2 in vitro, the expression of HO-1 in cells is further increased, reducing susceptibility to oxidative stress induced by FAC-induced iron overload. Our research has revealed new functions of HO-1 in managing liver iron overload and the potential regulatory pathway of Lnc286.2.

## 2. Materials and Methods

### 2.1. Animals and Treatment

Sixteen 7-week-old male Sprague Dawley rats were obtained from Nanjing Medical University and housed under standard conditions (22 ± 0.5 °C temperature, 50 ± 5% humidity, and 12 h light/dark cycle), with free access to distilled water and food. After a 1-week acclimatization period, the rats were randomly assigned to two groups, each consisting of eight rats: a control group (CON) and an iron overload group (IO). The CON group received vehicle injections every 3 days, while the IO group received iron dextran injections (150 mg/kg) every 3 days.

By the conclusion of the fourth week, the rats were subjected to an overnight fast prior to euthanasia. Blood samples were collected, and the plasma was processed and stored at −80 °C for subsequent analysis. A portion of the liver tissue was fixed in a 4% paraformaldehyde solution for subsequent hematoxylin and eosin (H&E) staining. The remaining liver samples were also stored at −80 °C. All animal procedures followed the guidelines set by the Animal Ethics Committee at Nanjing Agricultural University and the ’Guidelines on Ethical Treatment of Experimental Animals’ (2006) established by the Chinese Ministry of Science and Technology.

### 2.2. Determination of Liver Iron Content

A total of 0.5 g of liver tissue was measured out and subjected to digestion in accordance with the methods detailed in a previous study [22]. The concentration of total iron in the liver samples was determined using a Thermo iCE-3500 graphite atomic absorption spectrophotometer from Thermo Scientific, Wilmington, DE, USA, and the results are presented as micrograms per gram of wet tissue

### 2.3. Hematoxylin and Eosin (HE) Staining

Hematoxylin and eosin (H&E) staining was conducted according to previously described methods [23]. Fresh liver samples were promptly fixed in a 4% paraformaldehyde solution for at least 24 h, then embedded in paraffin and sectioned longitudinally at 5 µm thickness. Following deparaffinization with xylene, the sections underwent rehydration through a graded ethanol series (100%, 95%, 75%) prior to H&E staining. The stained sections were mounted with neutral resin and subsequently examined under a microscope for histopathological assessment and image analysis.

### 2.4. NAS Score

The following histologic data were analyzed: steatosis and inflammation, the score of each component of the NAS (steatosis (0–3 points) and lobular inflammation (0–3 points)). (1) Steatosis: 0 points (<5%); 1 point (5–33%); 2 points (34–66%); 3 points (>66%). (2) Lobular inflammation (counting necrotic foci): 0 points (none); 1 point (<2); 2 points (2–4); 3 points (>4) [24].

### 2.5. Quantitative Proteomics Transcriptomics Analysis

Tandem Mass Tags (TMT) labeling was utilized for quantitative proteomics analysis. Protein samples underwent trypsin digestion, TMT labeling, HPLC fractionation, and LC-MS/MS analysis. RNA samples were extracted for library preparation, sequencing, quality control, and read mapping. Data analysis was conducted according to the procedures outlined in the Appendix A. Shanghai Majorbio Co., Ltd. (Shanghai, China) supported the TMT proteomics and transcriptomics analyses.

### 2.6. Protein Extraction and Western Blot Assay

Protein concentrations were determined using the Easy II Protein Quantitative Kit (DQ111, TransGen Biotech Co., Beijing, China), following the manufacturer’s protocol. A total of 30 µg of protein per lane was subjected to electrophoresis on 12% sodium dodecyl sulfate–polyacrylamide gels and then transferred onto polyvinylidene fluoride membranes. The resulting images were captured with the VersaDoc 4000MP system (Bio-Rad, Hercules, CA, USA), and band densities were quantified using Quantity One 4.62 software (Bio-Rad, Hercules, CA, USA). Tubulin was employed as an internal loading control. Details regarding the primary and secondary antibodies used are provided in Appendix A.

### 2.7. RNA Isolation and Real-Time qPCR

Total RNA was extracted from liver samples and cultured cells using Trizol reagent, adhering to established protocols [25]. Cells were first washed with phosphate-buffered saline (PBS), followed by lysis in Trizol for RNA isolation via chloroform extraction, isopropanol precipitation, and ethanol washes. The purified RNA was then resuspended in diethyl pyrocarbonate-treated water. Complementary DNA (cDNA) was synthesized from 1 μg of total RNA using the Uni All-in-One First-Strand cDNA Synthesis SuperMix from TransGen Biotech, Beijing, China. Real-time quantitative PCR was conducted using a 1:20 dilution of cDNA on an Mx3000P Real-Time PCR System from Stratagene, CA, USA. Appendix A lists the primers, which were synthesized by Tsingke Bio-technology, Nanjing, China. The analysis employed the 2^−ΔΔCT^ method.

### 2.8. LncRNA Subcellular Location Prediction

LncRNA subcellular location was predicted by lncLocator (http://www.csbio.sjtu.edu.cn/bioinf/lncLocator/, accessed on 1 December 2023) from the Pattern Recognition and Bioinformatics Group of Shanghai Jiao Tong University.

### 2.9. Cell Culture and Drug Treatment

Rat liver-derived fibroblast-like cell lines (BRL-3A) were procured from Beijing BeNa Culture Collection Co. Ltd. (Beijing, China) and cultured in a controlled environment with 5% CO_2_ at 37 °C. The cells were maintained in Dulbecco’s Modified Eagle’s Medium (DMEM) (319-005-CL, Wisent, QC, Canada), supplemented with 10% fetal bovine serum (FSP500, ExCell Bio, Suzhou, China) and 100 IU/mL of penicillin/streptomycin (FG101-01, TransGen Biotech, Beijing, China). Ferric ammonium citrate (FAC) (F5879, Sigma-Aldrich, St. Louis, MO, USA) was used in the experiments. Cobaltic protoporphyrin IX chloride (CoPP) (sc294098, Santa Cruz Biotechnology, Shanghai, China) was resuspended in dimethyl sulfoxide (DMSO) to achieve a final concentration of less than 1%. Control cells were treated with the respective vehicle controls.

### 2.10. Cell Viability Assay

A total of 1 × 10^4^ cells were seeded into each well of 48-well plates and subsequently treated with ferric ammonium citrate (FAC) and cobaltic protoporphyrin IX chloride (CoPP). Cell viability and proliferation were assessed using the Cell Counting Kit-8 (CCK-8) assay (K1018, ApexBio Technology, Shanghai, China). After treatment durations of 12 or 24 h, 25 μL of the CCK-8 reagent was added to each well, followed by a 1 h incubation at 37 °C. Absorbance at 450 nm was measured using a microplate reader (Synergy H1, BioTek, Winooski, VT, USA).

### 2.11. Live/Dead Cell Staining

Post-treatment, the cells were rinsed with phosphate-buffered saline (PBS) and then incubated with 2 μM of Calcein-AM and 4.5 μM of propidium iodide (PI) (C542, Beijing Tongren Institute of Chemistry, Beijing, China) for 15 min at 37 °C. Confocal laser scanning microscopy was employed for image acquisition. Subsequently, the proportions of viable cells (stained with Calcein-AM, negative for PI) and non-viable cells (negative for Calcein-AM, positive for PI) were quantified.

### 2.12. Assessment of Fe^2+^, ROS, and Lipid Peroxidation by Flow Cytometry

Cellular ferrous iron (Fe^2+^) content was assessed by incubating cells with 1 mM of FerroOrange (F374, Beijing Tongren Institute of Chemistry, Beijing, China) for 30 min at 37 °C in the dark, followed by immediate flow cytometry analysis. Fluorescence intensity was measured in the PE-A channel. ROS levels were evaluated by incubating cells with 5 μM of CellROX^®^ Green Reagent (C10444, Invitrogen, Carlsbad, CA, USA) for 30 min at 37 °C in the dark. Subsequently, cells were collected, rinsed with Hank’s Balanced Salt Solution, and analyzed using flow cytometry. Lipid peroxidation was assessed using Liperfluo staining (L248, Beijing Tongren Institute of Chemistry, Beijing, China). After a 30 min staining with Liperfluo at 37 °C, cells were trypsinized, and flow cytometry analysis was performed immediately. The fluorescence intensity for both ROS and lipid peroxidation was monitored in the FITC channel.

### 2.13. Glutathione (GSH) and Malondialdehyde (MDA) Assay

The levels of glutathione (GSH) and malondialdehyde (MDA) were quantified using respective assay kits (A061-2-1 for GSH and A003-1-2 for MDA; JianCheng, Nanjing, China), in accordance with the manufacturer’s guidelines. A microplate reader (Synergy H1, BioTek, USA) was employed to measure the absorbance.

### 2.14. Immunofluorescence

Immunofluorescence (IF) assays were conducted to detect endogenous and transfected proteins within cell lines. Cells cultured on glass slides were fixed with 4% paraformaldehyde, permeabilized with Triton X-100, blocked with bovine serum albumin, and then incubated with HO-1 primary antibodies. After washing, the slides were incubated with goat anti-rabbit IgG (H + L) secondary antibodies. Nuclei were counterstained with 4′,6-diamidino-2-phenylindole (DAPI). The assembled slides were examined using a fluorescence microscope from Leica Microsystems, Wetzlar, Germany, specifically the DMI6000 B model.

### 2.15. Transfection Procedures

BRL-3A cell lines were transfected with small interfering RNAs (siRNAs) at a concentration of 50 nM for knockdown or plasmids at a concentration of 50 nM for overexpression using the jetPRIME transfection reagent (114–15, Polyplus Transfection, Illkirch, France) over a 12 h period. Subsequently, the transfected cells were treated with FAC or phosphate-buffered saline (PBS) for an additional 24 h. The efficacy of siRNA-mediated knockdown and plasmid-mediated upregulation was validated through qRT-PCR and Western blot analysis. The siRNA sequences used are provided in the text. Ho-1: siRNA (5′-CCGUGGCAGUGGGAAUUUATT-3′); Lnc286.2: siRNA (5′-CAUAAAGUCCAUAAAUACAUU-3′).

### 2.16. Datasets

Data were collected from a minimum of three independent biological replicates and are presented as means ± standard error of the mean (SEM). GraphPad Prism 9.0 software was employed to analyze mean differences. A one-way analysis of variance (ANOVA) was used for multiple group comparisons, while *t*-tests were applied for comparisons between two groups. Statistical significance was determined at a *p*-value threshold of less than 0.05.

## 3. Results

### 3.1. Proteomic Analysis Indicates Elevated HO-1 Expression in Rat Liver Due to Iron Overload

The process of integrated analysis of proteomics and transcriptomics is shown in Figure 1A. Iron overload increased hepatic iron deposition in rats (Figure 1B, *p* < 0.01). Meanwhile, iron overload-induced steatosis and inflammation in rats (Figure 1C,D, *p* < 0.05). To thoroughly elucidate the impact of iron overload on hepatic protein expression, we performed a comprehensive analysis of the proteomic alterations in rat livers subjected to iron overload using TMT-based proteomic analysis techniques. Proteomic profiling facilitated by TMT technology revealed 647 proteins exhibiting differential expression in rat liver tissues after a four-week period of iron supplementation, both with and without induced iron overload conditions. As illustrated by the PCA (Principal Component Analysis) plot results, a clear separation can be observed between the samples of the two groups (Appendix A). In particular, the treatment with iron overload in rats led to the upregulation of 477 proteins and the downregulation of 170 proteins in the liver (Figure 1E). An analysis was conducted to evaluate the changes in mRNA expression within the liver tissues of rats that underwent iron overload treatment. The PCA score clearly distinguished the iron overload group from the control group (Appendix A). A total of 977 mRNAs were upregulated, and 358 mRNAs were downregulated (Figure 1F). The top 50 mRNAs that were found to be upregulated or downregulated were used to construct a heatmap (Appendix A). The intersection of all differentially expressed proteins and all differentially expressed mRNAs was removed, resulting in a total of 178 (Figure 1G). Out of these 178, 147 were upregulated proteins, and 31 were downregulated proteins; the same applies to mRNA (Figure 1H). To identify key proteins in the iron overload-induced rat liver injury model, The top 50 proteins that were found to be upregulated and the top 50 proteins that were found to be downregulated were selected for the purpose of constructing a heatmap (Appendix A). GO and KEGG enrichment analyses were then performed on these proteins (Appendix A). In further analysis, a protein-protein interaction network was constructed using these proteins. As shown in Figure 1I, HO-1 occupied a central position in the network, suggesting its importance in the iron overload-induced rat liver injury model. To identify proteins closely associated with HO-1 in this network, we selected the top 10 proteins using MCC scores, as shown in Figure 1J. The protein expression and transcriptomic sequencing results of HO-1 are shown in Figure 1K and Figure 1M, respectively. We verified their expression using Western blot and RT-qPCR, and the results are consistent with the omics data, as shown in Figure 1L, N, original western blot figures can be found in Appendix A.

### 3.2. Transcriptomics Analysis Reveals Ho-1 Is Targeted by Differentially Expressed LncRNA in Rat Liver

Next, we will attempt to identify potential LncRNAs that regulate HO-1. The PCA plot revealed a significant separation between the iron overload group and the control group (Appendix A). In the rat liver injury model induced by iron overload, the analysis revealed 138 differentially expressed LncRNAs, comprising 64 upregulated and 74 downregulated LncRNAs (Figure 2A). The heatmap displays the expression levels of the top 10 upregulated and top 10 downregulated LncRNAs (Figure 2B). The heatmap of the top 50 upregulated and top 50 downregulated LncRNAs can be found in Appendix A. Figure 2C displays all differentially expressed LncRNAs targeting *Ho-1*. Due to the consistent changes in protein and mRNA levels of HO-1, both of which are upregulated in iron overload-induced liver injury in rats, we conducted subcellular localization analysis of all LncRNAs in Figure 2C and selected two nuclear-expressed LncRNAs (Figure 2D) as potential candidates for regulating *Ho-1*. The transcriptomic results of Lnc286.2 and Lnc362.2, as well as the RT-qPCR results, are shown in Figure 1E (*p* < 0.05; *p* < 0.01) and Figure 1F (*p* < 0.01), respectively. Both demonstrate consistent downregulation in liver injury induced by iron overload in rats.

### 3.3. FAC Induces Iron Overload and High Expression Level of HO-1 in BRL-3A Cells

The iron overload model in BRL-3A cells was established using FAC treatment for 12 and 24 h, demonstrating a time- and dose-dependent increase in iron overload (Figure 3A,B, *p* < 0.01). Bright-field microscopy revealed decreased viability in BRL-3A cells exposed to FAC at both 12 and 24 h (Figure 3C) or live/dead staining (Figure 3D). The intracellular levels of Fe^2+^ (Figure 3E, *p* < 0.05), ROS (Figure 3F, *p* < 0.01), lipid ROS (Figure 3G, *p* < 0.01), and MDA (Figure 3I, *p* < 0.01) were increased in BRL-3A cells treated with FAC, accompanied by GSH depletion (Figure 3H, *p* < 0.05). The data indicates that FAC induces iron overload in BRL-3A cells. Meanwhile, the RT-qPCR (Figure 3J, *p* < 0.01), Western blot (Figure 3K, *p* < 0.01), and immunofluorescence assay (Figure 3L) confirmed that FAC increased HO-1 expression in BRL-3A cells, which was consistent with proteomics and transcriptomics, original western blot figures can be found in Appendix A.

### 3.4. HO-1 Knockdown Promotes Iron Overload in FAC-Triggered BRL-3A Cells

RT-qPCR (Figure 4A, *p* < 0.01), Western blot (Figure 4B, *p* < 0.01), and immunofluorescence (Figure 4C) showed that transfection with siHo-1 diminished FAC-induced HO-1 improvement, original western blot figures can be found in Appendix A. Transfection with siHo-1 prior to FAC treatment led to a significant exacerbation of the FAC-induced decrease in cell viability (Figure 4D, *p* < 0.01) and a reduction in the viable cell count (Figure 4E). siHo-1 transfection significantly reduced FAC-induced increases in ROS (Figure 4G, *p* < 0.05), lipid ROS (Figure 4H, *p* < 0.01), and MDA (Figure 4J, *p* < 0.05) levels while mitigating the decrease in GSH levels (Figure 4I, *p* < 0.01) in BRL-3A cells. However, the FAC-induced changes in Fe^2+^ content could not be aggravated in BRL-3A cells transfected by siHo-1 (Figure 4F, *p* > 0.05). These results indicated that HO-1 inhibition deteriorated iron overload in FAC-induced BRL-3A cells by increasing ROS, lipid ROS, and MDA production while decreasing GSH production.

### 3.5. Promoting HO-1 Helps Reduce FAC-Triggered Iron Overload in BRL-3A Cells

To substantiate the role of HO-1 in the context of FAC-induced iron overload in BRL-3A cells, we utilized CoPP to enhance HO-1 expression. The cells were incubated with CoPP for 12 and 24 h, after which their viability was assessed (Figure 5A,B). A concentration of 50 μM of CoPP was employed to pre-treat BRL-3A cells for 12 h (Figure 5C, *p* < 0.01) and 24 h (Figure 5D, *p* < 0.05) prior to FAC exposure, resulting in a preservation of cell viability. Concurrently, Western blot analysis (Figure 5E, *p* < 0.05) and immunofluorescence assays (Figure 5F) verified that treatment with CoPP for 12 and 24 h elevated HO-1 expression in BRL-3A cells, original western blot figures can be found in Appendix A. We employed 50 µM of CoPP to pretreat BRL-3A cells for 12 h before exposure to FAC; an increased number of living cells (Figure 5G) was observed. The RT-qPCR (Figure 5H, *p* < 0.01), Western blot (Figure 5I, *p* < 0.01), and immunofluorescence assay (Figure 5J) validated the enhancement of FAC-induced HO-1 upregulation in BRL-3A cells following a 12 h pretreatment with CoPP, original western blot figures can be found in Appendix A. Moreover, this pretreatment with CoPP mitigated the FAC-induced increase in ROS (Figure 5L, *p* < 0.01) and lipid ROS (Figure 5M, *p* < 0.01) levels and counteracted the decrease in GSH levels (Figure 5N, *p* < 0.05). However, the elevation of Fe^2+^ and MDA content induced by FAC was not significantly affected in BRL-3A cells pretreated with CoPP (Figure 5K and 5O, *p* > 0.05). These findings suggest that the upregulation of HO-1 ameliorates iron overload in BRL-3A cells challenged with FAC by decreasing ROS and lipid ROS production and enhancing GSH synthesis.

### 3.6. Lnc286.2 Inhibition Promotes HO-1 Expression in FAC-Triggered Iron Overload in BRL-3A Cell

RT-qPCR revealed that FAC downregulated Lnc286.2 expression (Figure 6A, *p* < 0.01) in BRL-3A cells, which was consistent with transcriptomics. In order to explore whether Lnc286.2 is regulating HO-1 or not, we employed siRNA to knockdown Lnc286.2 expression. Analysis via RT-qPCR (Figure 6B, *p* < 0.05), Western blotting (Figure 6C, *p* < 0.05), and immunofluorescence (Figure 6D) revealed that siLnc286.2 transfection potentiated the upregulation of HO-1 induced by FAC treatment, original western blot figures can be found in Appendix A. Transfection with siLnc286.2 prior to FAC exposure markedly counteracted the FAC-induced reduction in cellular viability (Figure 6E, *p* < 0.01) and the decrease in the viable cell count (Figure 6F). Furthermore, the transfection of siLnc286.2 also mitigated the FAC-triggered increase in lipid ROS levels in BRL-3A cells (Figure 6H, *p* < 0.05). However, the FAC-induced enhancement in ROS content could not be aggravated in BRL-3A cells transfected by siLnc286.2 (Figure 6G, *p* > 0.05). These results indicated that Lnc286.2 inhibition promoted HO-1 expression in FAC-induced BRL-3A cells and reversed the production of lipid ROS.

## 4. Discussion

Iron plays an essential role in animal physiology. On the one hand, it is essential for heme biosynthesis, oxygen transport, DNA biosynthesis, and the citric acid cycle [26]. On the other hand, excess iron is toxic because it produces ROS through the Fenton reaction, which leads to severe cellular dysfunction and organ damage [27]. Considering the crucial role of the liver in iron metabolism, it rapidly exhibits pathological changes in cases of iron overload. Unbound iron facilitates the generation of ROS, consequently inducing lipid peroxidation. Elevated iron levels in the liver stimulate lipid peroxidation, thereby contributing to the advancement of non-alcoholic fatty liver disease (NAFLD), cirrhosis, or hepatocellular carcinoma [28,29]. Wang et al. established an iron overload model in mice by administering 50mg/kg of dextran iron intraperitoneally for 7 weeks. H&E staining revealed abundant yellow-brown iron deposits in the liver tissue, along with inflammatory cell infiltration and increased liver fibrosis area [30]. Current research has demonstrated that iron toxicity, either alone or in synergy with other hepatotoxins, such as alcohol and thioacetamide, can cause damage and fibrosis through oxidative stress pathways [31]. Reactive oxygen species (ROS) and iron may participate in liver fibrosis in two ways: (1) as direct inducers of necrosis in liver parenchymal cells and (2) as activators of cells responsible for producing fibrogenic mediators [32]. This is similar to the chronic liver injury induced by thioacetamide injections, which activates the fibrotic process in the liver, including the induction of proliferation of hepatic stellate cells—the primary source of myofibroblasts—that produce large amounts of extracellular matrix, leading to the formation of fibrotic scars [33]. Our study used a dose of 150 mg/kg of dextran iron administered intraperitoneally to establish an iron overload model in rats. H&E staining results indicated that iron overload led to significant vacuolization and lipid degeneration in rat liver cells, accompanied by increased inflammatory cells and visible brown iron deposits. This suggests that iron overload causes evident damage to the liver.

Through the analysis of proteomics, we identified the key protein HO-1 in the rat liver injury model induced by iron overload. In addition, autophagy-related proteins CTSD and CTSB, iron metabolism-related proteins FTH1, FTL1, HAMP, TFRC, and redox reaction-related proteins CYP1A2, GSTP1, and AKR7A3 are also in key positions. The HO family includes three types of heme oxygenases; HO-1, -2, and -3 are distinct isoforms, each encoded by separate genes and governed by unique regulatory mechanisms [34]. HO-1, classified as a heat shock protein, is encoded by the Hmox1 gene and is ubiquitously expressed across various cell types. Typically, under normal physiological conditions, HO-1 exhibits minimal expression in the majority of tissues; however, its expression can be significantly upregulated in response to a myriad of pathological and physiological stressors. Consequently, the upregulation of HO-1 serves as an indicator of the adaptive protective response mounted by cells and tissues when confronted with external stress stimuli [35]. HO-1 breaks down hemoglobin into carbon monoxide, ferrous ions, and biliverdin [34,36]. CO possesses extensive therapeutic capabilities, acting as both an anti-inflammatory and neuroprotective agent, characterized by potent immunomodulatory properties and stability under physiological conditions [37]. Ferrous ions trigger the upregulation of ferritin heavy chain expression and enhance the antioxidant properties of HO-1 [38]. Biliverdin is metabolized into bilirubin by the action of biliverdin reductase, thereby fulfilling a protective role at the cellular level. Moreover, both biliverdin and bilirubin are recognized as active scavengers of reactive oxygen species, exhibiting potential anti-inflammatory and antioxidant activities [39]. Therefore, HO-1 and heme metabolites are considered to have antioxidant and cellular protective effects.

Research indicates that elevated levels of HO-1 contribute to the alleviation of oxidative stress. Adedoyin et al. have shown that HO-1 mitigates cell death triggered by iron overload in renal epithelial cells [40]. Park et al. have demonstrated that Schisandrin A diminishes NF-κB signaling, stimulates the expression of HO-1, and suppresses the expression of tumor necrosis factor-alpha (TNF-α), interleukin-1 beta (IL-1β), and IL-6, consequently conferring an anti-inflammatory response in endotoxin-stimulated RAW 264.7 cells [41]. Nevertheless, persistent overexpression of HO-1 may also initiate ferroptosis. In a pioneering study, Chang et al. identified that Bay, an inhibitor of IκBα, enhances ferroptosis via the Nrf2-SLC7A11-HO-1 signaling cascade [9]. Kwon et al. have demonstrated that the expression of HO-1 is essential for cell ferroptosis triggered by erastin [42]. These findings suggest that the dual function of HO-1 in iron overload could be contingent upon various pathological and physiological states. The protective effect of HO-1 on cells partially relies on reducing oxidative stress levels. Gabunia et al. have discovered that IL-19 can mitigate ROS levels by enhancing the expression of HO-1 in human vascular smooth muscle cells [43]. Lu et al. showed that the decrease in HO-1 expression and activity in D5 dopamine receptor knockout mice led to an increase in ROS [44]. This study found that HO-1 inhibition exacerbates oxidative stress levels in BRL-3A cells, increases lipid peroxidation, and promotes cell death caused by iron overload. Conversely, HO-1 overexpression significantly alleviates the increase in oxidative stress levels and mitigates cell death in BRL-3A cells induced by iron overload. LncRNAs constitute a category of non-coding RNA molecules exceeding 200 nucleotides in length. They do not have the ability to encode proteins, but they play important regulatory roles in gene expression, chromatin remodeling, and cell cycle regulation [45]. LncRNAs with heterogeneous expression levels have regulatory functions at multiple levels, from transcription to protein modification [46]. LncRNA can regulate gene transcription activity by interacting with DNA sequences, affecting cellular gene expression patterns [47]. Furthermore, it has the capability to interact with splicing factors and mRNA, thereby modulating RNA splicing and mRNA stability, which in turn affects the patterns of gene expression [48]. Research has indicated that LncRNAs are pivotal in the pathogenesis of liver diseases. Specifically, Yuan and colleagues have discovered that LncRNA TLNC1 facilitates the proliferation and spread of hepatocellular carcinoma by suppressing the p53 signaling pathway [49]. Chen and colleagues have demonstrated that the long non-coding RNA known as Airn can mitigate the progression of liver fibrosis by modulating the KLF2-eNOS-sGC signaling pathway [50]. Lnc286.2 is the only LncRNA that shows consistent expression patterns in transcriptomic sequencing, liver, and cellular expression trends. It targets Ho-1, and its subcellular localization is predicted to be in the nucleus. Since Lnc286.2 is downregulated in the iron overload model while HO-1 is upregulated, and their protein and mRNA expression trends align, we believe that Lnc286.2 acts as an inhibitor of HO-1, regulating its transcription. Our study found that inhibiting Lnc286.2 can increase HO-1 expression, alleviate decreased cell viability in BRL-3A cells, and mitigate increased lipid ROS production within the cells. However, it does not have an effect on overall ROS production increase. While siLnc286.2 can enhance HO-1 expression, its effect is not as strong as CoPP, leading us to speculate that siLnc286.2’s ability to enhance HO-1 may not be sufficient to counteract the increase in ROS levels.

## 5. Conclusions

A preliminary investigation was conducted into the protective effects of heme oxygenase-1 (HO-1) in a liver injury model induced by iron overload. In vivo experiments revealed that HO-1 was centrally located within the protein-protein interaction network formed by the top 50 upregulated and top 50 downregulated differential proteins. In addition, the validation of HO-1 expression levels through Western blot analysis has further substantiated its potential as a vital factor in this process. Furthermore, transcriptomic analysis of iron-overloaded rat livers has identified the potential regulatory relationship of Lnc286.2 and HO-1. Subsequent in vitro experiments established an iron overload model using BRL-3A cells and verified the regulatory effects of HO-1 and Lnc286.2.

In summary, the results of this study demonstrate that Lnc286.2 has the capacity to suppress HO-1 expression and that the knockdown of Lnc286.2 results in the promotion of HO-1 expression. This, in turn, has a beneficial effect on the level of lipid peroxidation within cells, enhances antioxidant capacity, and mitigates liver cell injury induced by iron overload.

## Figures and Tables

**Figure 1 biology-14-00081-f001:**
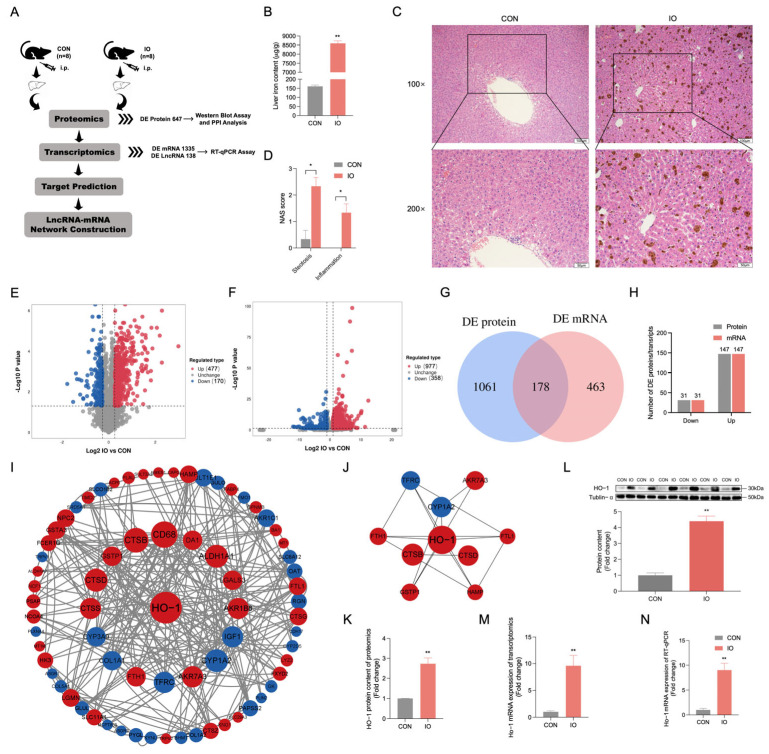
HO-1 is crucial for mediating the liver damage induced by iron overload in rats. (**A**) Flow chart of analysis. (**B**) Hepatic iron deposition. (**C**) Liver H&E staining. (**D**) NAS score. (**E**) Differentially expressed proteins are displayed as a volcano plot. (**F**) Differentially expressed mRNAs are displayed as a volcano plot. (**G**) Venn diagram of differential expression proteins and differential expression mRNAs. (**H**) Statistics of proteins and mRNAs expression at the intersection of Venn diagram. (**I**) Protein–protein interaction network of 50 most differentially upregulated proteins and 50 most differentially downregulated proteins, blue represents downregulation, red represents upregulation, the size of the circle represents degree number, and the line between the circle represents an interaction. (**J**) MCC top 10 proteins of the protein-protein interaction network, blue represents downregulation, red represents upregulation, the size of the circle represents degree value, and the line between the circle represents interaction. (**K**) The expression of the HO-1 protein was shown as a histogram (n = 4). (**L**) Western blot was used to detect HO-1 protein content (n = 6). (**M**) The expression of HO-1 mRNA was shown as a histogram (n = 3). (**N**) RT-qPCR was used to detect the HO-1 mRNA expression (n = 6). CON: control, IO: iron overload. Values are means ± SEM, * *p* < 0.05, ** *p* < 0.01.

**Figure 2 biology-14-00081-f002:**
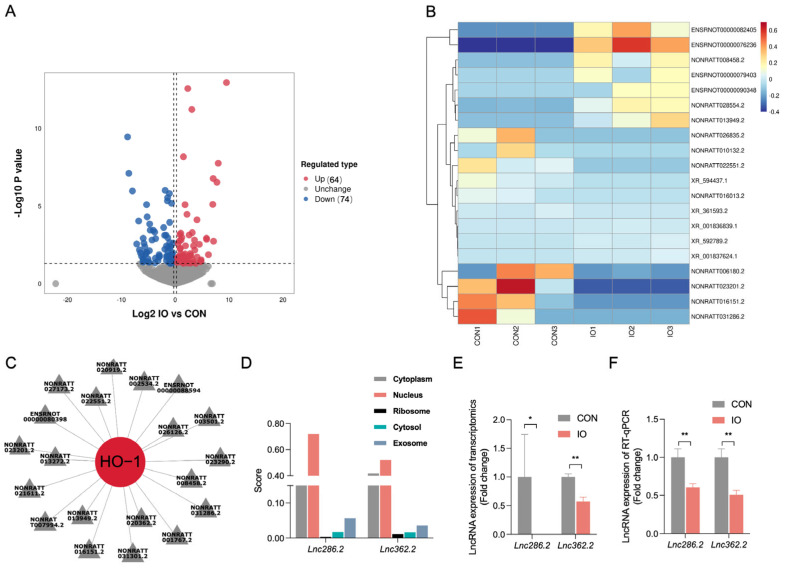
LncRNAs that potentially regulate HO-1. (**A**) Volcano plot analysis for the differentially expressed LncRNAs. (**B**) Heatmap representing hierarchical clustering of 10 most differentially upregulated LncRNAs and 10 most differentially downregulated LncRNAs. (**C**) Differentially expressed LncRNAs targeted to *Ho-1*. (**D**) LncRNA subcellular location prediction. (**E**) Expression of LncRNA is shown as a histogram (n = 3). (**F**) RT-qPCR was used to detect LncRNA expression (n = 6). CON: control, IO: iron overload. Values are means ± SEM, * *p* < 0.05, ** *p* < 0.01.

**Figure 3 biology-14-00081-f003:**
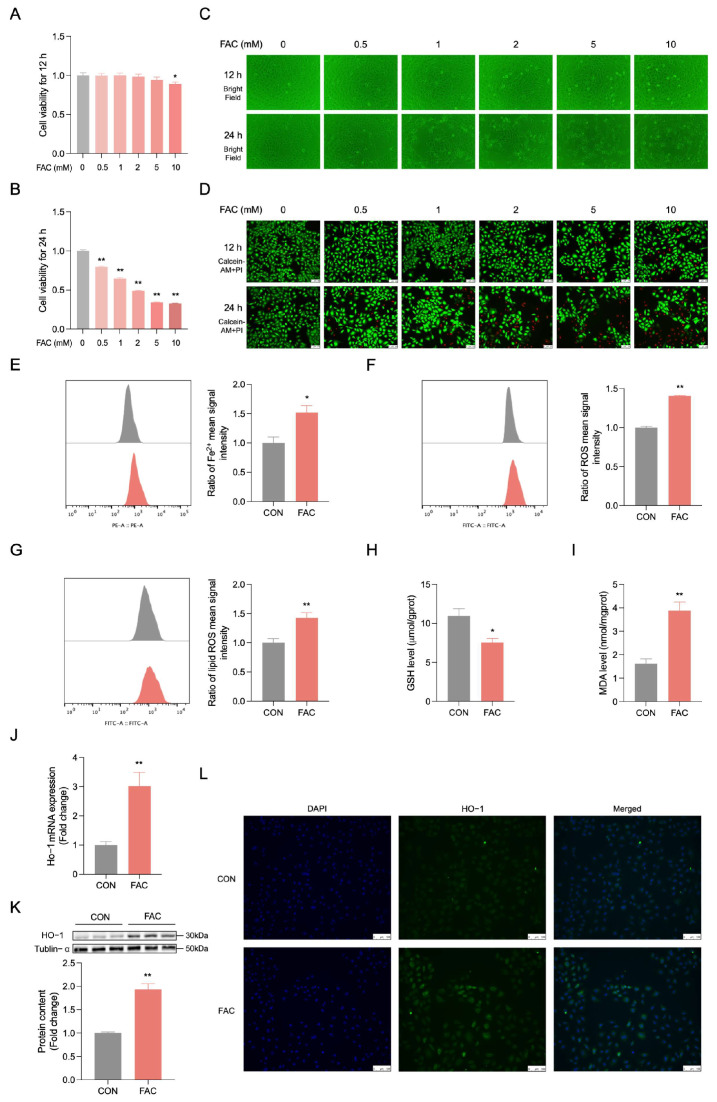
FAC induces iron overload in BRL-3A cells. (**A**,**B**) BRL-3A cells were subjected to treatment with varying concentrations of FAC (0, 0.5, 1, 2, 5, 10 mM) for either 12 or 24 h. The cell viability was assessed using the CCK-8 kit with four replicates (n = 4). (**C**) The representative changes in cell morphology are depicted. (**D**) Live and dead cell staining: Calcein-AM staining (green) represents live cells, while PI staining (red) represents dead cells. scale bar: 100 μM. (**E**) The intracellular Fe^2+^ was estimated by FerroOrange staining and analyzed by flow cytometry after being treated with 2 mM of FAC at 24 h (n = 3). (**F**) Intracellular ROS levels were measured using flow cytometry after being treated with 2 mM of FAC at 24 h (n = 3). (**G**) Intracellular lipid ROS levels were estimated by Liperfluo staining and analyzed by flow cytometry after being treated with 2 mM of FAC at 24 h (n = 3). (**H**,**I**) After exposure to 2 mM of FAC for a duration of 24 h, the levels of intracellular GSH and MDA were quantified using a GSH/MDA assay kit with triplicate measurements (n = 3). (**J**) The expression of Ho-1 mRNA was determined through reverse transcription-quantitative polymerase chain reaction (RT-qPCR), also with triplicate samples (n = 3). (**K**) The expression of HO-1 protein in BRL-3A cells was analyzed by Western blotting with three replicates (n = 3). (**L**) Immunofluorescence microscopy was employed to visualize the co-expression of HO-1 (fluorescing green) alongside DAPI nuclear staining (appearing blue) at 40× magnification (n = 3), scale bar: 100 μM. CON: control. Values are means ± SEM, * *p* < 0.05, ** *p* < 0.01.

**Figure 4 biology-14-00081-f004:**
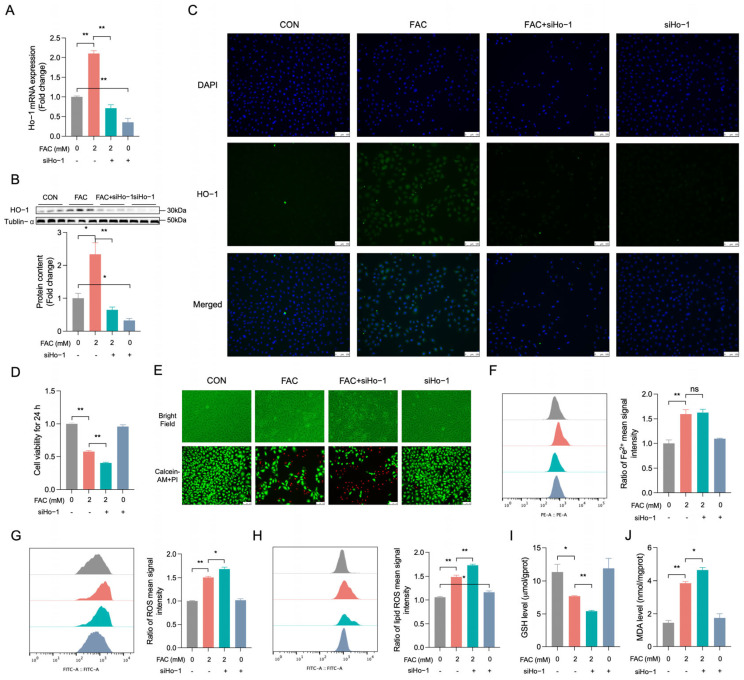
Suppressing HO-1 intensified FAC-induced injury in BRL-3A cells. (**A**) RT-qPCR was employed to quantify the expression levels of Ho-1 mRNA (n = 3). (**B**) Western blotting was conducted to assess the expression levels of the HO-1 protein within BRL-3A cells (n = 3). (**C**) Immunofluorescence microscopy images show the co-expression of HO-1 (green) with DAPI (blue) at 40× magnification (n = 3), scale bar: 100 μM. (**D**) Cell viability was assayed by a CCK-8 kit (n = 4). (**E**) Illustrated are the characteristic morphological alterations in the cells (upper panel); Calcein-AM staining (green) represents live cells, while PI staining (red) represents dead cells (down). (**F**) Intracellular Fe^2+^ was estimated by FerroOrange staining and analyzed by flow cytometry (n = 3). (**G**) The levels of intracellular ROS were quantified employing flow cytometry (n = 3). (**H**) Intracellular lipid ROS levels were estimated by Liperfluo staining and analyzed by flow cytometry (n = 3). (**I**,**J**) Intracellular GSH/MDA were measured by GSH/MDA assay kit (n = 3). CON: control, FAC + siHo-1: pre-transfected siHo-1 for 12 h + 2 mM FAC, siHo-1: transfected siHo-1. Values are means ± SEM, * *p* < 0.05, ** *p* < 0.01, and “ns” indicates no significant difference.

**Figure 5 biology-14-00081-f005:**
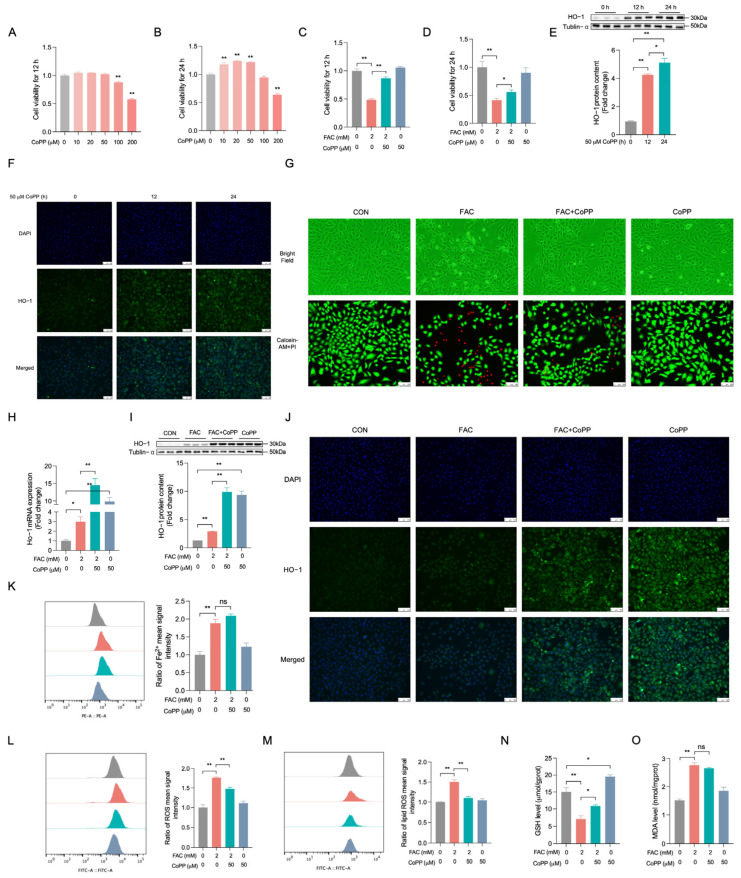
HO-1 overexpression alleviates FAC-induced injury in BRL-3A cells. (**A**,**B**) The cells were exposed to varying concentrations of CoPP (0, 10, 20, 50, 100, 200 μM) for either 12 or 24 h, and their viability was assessed using the CCK-8 assay kit (n = 4). (**C**,**D**) Prior to FAC treatment, BRL-3A cells were pre-incubated with 50 μM of CoPP for 12 or 24 h, after which cell viability was again evaluated using the CCK-8 assay kit (n = 4). (**E**) Western blot analysis was conducted to quantify HO-1 protein expression in BRL-3A cells (n = 3). (**F**) Immunofluorescence microscopy was utilized to visualize the co-localization of HO-1 (green) with nuclear DAPI staining (blue) at a 40× magnification; (n = 3), with a scale bar indicating 100 μM. (**G**) Photographs depict cellular morphological alterations; live cells are indicated by Calcein-AM fluorescence (green), and dead cells are marked by propidium iodide (PI) fluorescence (red). (**H**) RT-qPCR was employed to measure HO-1 mRNA expression (n = 3). (**I**) Western blot analysis was repeated to confirm HO-1 protein expression in BRL-3A cells (n = 3). (**J**) Additional immunofluorescence microscopy images showed the co-expression of HO-1 (green) with DAPI (blue) at a 40× magnification (n = 3), with a scale bar of 100 μM. (**K**) Intracellular Fe^2+^ levels were determined using FerroOrange staining and flow cytometry analysis (n = 3). (**L**) Intracellular ROS levels were quantified using flow cytometry (n = 3). (**M**) Intracellular lipid ROS levels were estimated by Liperfluo staining and analyzed via flow cytometry (n = 3). (**N**,**O**) The levels of intracellular GSH and MDA were assessed using a GSH/MDA assay kit (n = 3). CON: control, FAC + CoPP: CoPP was pre-treated for 12 h before FAC treatment, CoPP: treated with CoPP. Values are means ± SEM, * *p* < 0.05, ** *p* < 0.01, and “ns” indicates no significant difference.

**Figure 6 biology-14-00081-f006:**
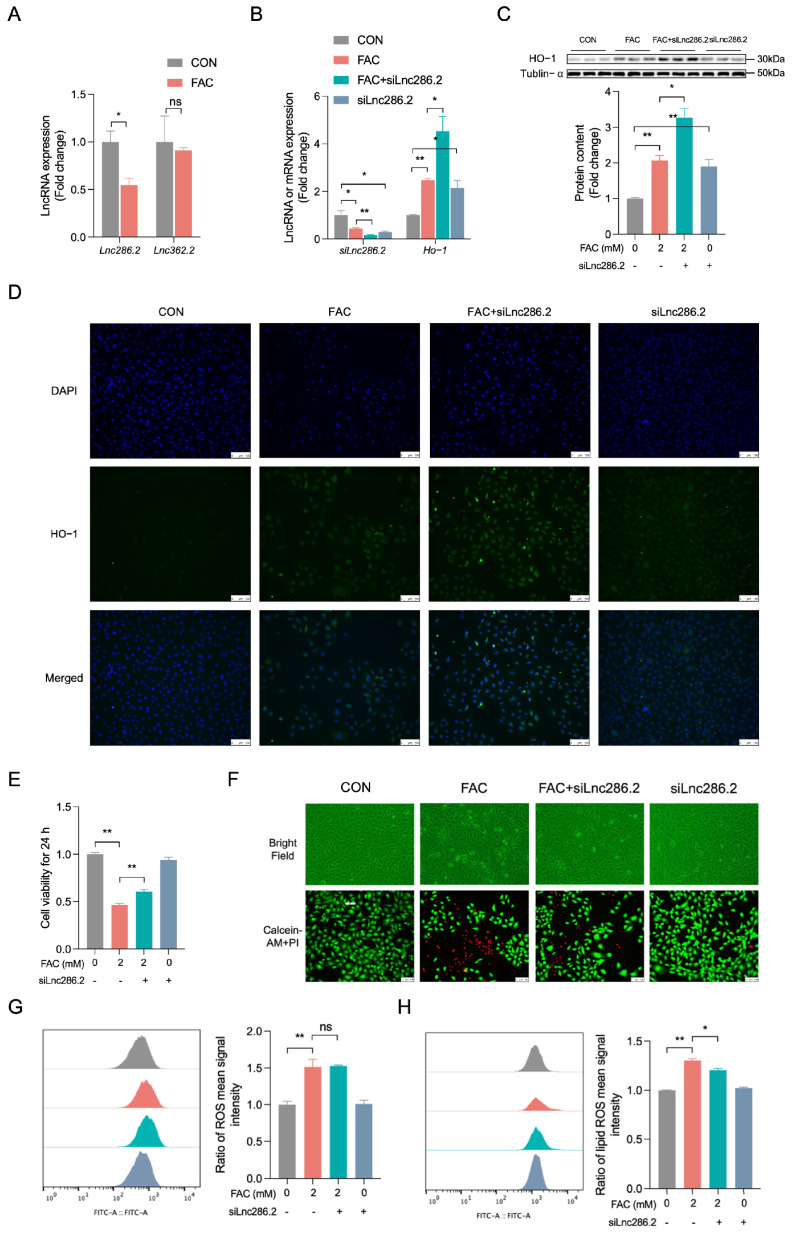
Lnc286.2 knockdown promotes HO-1 expression in FAC-treated BRL-3A cells. (**A**) RT-qPCR was employed to quantify the expression levels of Lnc286.2 and Lnc362.2 (n = 3). (**B**) RT-qPCR was similarly utilized to assess the expression of Lnc286.2 and HO-1 mRNA (n = 3). (**C**) Western blot analysis was conducted to determine the expression of HO-1 protein in BRL-3A cells (n = 3). (**D**) Immunofluorescence microscopy was used to visualize the co-localization of HO-1 (fluorescing green) with DAPI nuclear staining (appearing blue) at a 40× magnification (n = 3), with a scale bar indicating 100 μM. (**E**) Cellular viability was evaluated using the CCK-8 assay kit (n = 4). (**F**) Photographs depicting cellular morphological changes are presented above; live cells are indicated by Calcein-AM staining (green), while dead cells are marked by propidium iodide (PI) staining (red), as shown below. scale bar: 100 μM. (**G**) Intracellular levels of ROS were quantified by flow cytometry (n = 3). (**H**) Intracellular lipid ROS levels were estimated by Liperfluo staining and analyzed by flow cytometry (n = 3). CON: control, FAC + siLnc286.2: siLnc286.2 was pre-treated for 12 h before FAC treatment, siLnc286.2: treated with siLnc286.2. Values are means ± SEM, * *p* < 0.05, ** *p* < 0.01, and “ns” indicates no significant difference.

## Data Availability

The original contributions presented in this study are included in the article/Appendix A. Further inquiries can be directed to the corresponding authors.

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
