# Peer review of "Exploring the Mechanisms of Iron Overload-Induced Liver Injury in Rats Based on Transcriptomics and Proteomics"

_biology, 2025, doi:10.3390/biology14010081_

Round 1

Reviewer 1 Report

Comments and Suggestions for Authors

The manuscript describes the induction of HMOX1 by iron, and suggests a role of Lnc286.2 in the process. However, many results described in the manuscript have already been published. The induction of Hmox1 expression by iron has been known for a long time (see for example McDonald CJ et al, Am J Physiol Gastrointest Liver Physiol. 2011 Apr;300(4):G554-60. It is also very well known that iron overload causes liver damage, so the reported results regarding the in vivo effect on liver are not very surprising, given the extremely high dose of iron used. The introduction promises to reveal new functions of HMOX1 in iron overload (line 107), but no new functions for HMOX1 are mentioned in the Discussion. Also, it is not clear what prompted the authors to search for LncRNA involvement, given the general acceptance of HMOX1 induction by oxidative stress. Taking into account the limited amount of new data, the manuscript is far too long.

Major points:

Line 31: The abstract mentions “dextroglycoside”, while the rest of the manuscript mentions iron dextran.

Line 41: The abstract mentions “cobalt chloride” rather than cobalt protoporphyrin.

Line 71 and 77:  Introduction states that the study identifies HMOX1 as a mediator of ferroptosis; however, no new evidence for this statement is presented. Line 77 states that the study describes a role for FGF21; however, FGF21 is not further mentioned anywhere in the text.

Line 176: BRL-3A should be described as a liver-derived fibroblast-like cell line, rather than as a “liver cell”.

Regarding the immunoblots presented in the Supplementary Materials, please show the whole membrane – this will the reader allow to judge the specificity of the antibody. Also, include the molecular weight of the markers shown, as well as the marker manufacturer and ordering number.

Minor point:

Lines 90 to 93 state the same fact twice.

Reviewer 2 Report

Comments and Suggestions for Authors

It seems that most of the results are devoted to the study of the role of heme oxygenase. In my opinion, the title of the work should be written in accordance with the obtained results

1) The purpose of the study By combining transcriptomics and proteomics analysis, we investigate the mechanisms underlying iron overload-induced liver damage in rats. In this case, the authors list the obtained results in the abstract, but do not indicate the studied mechanism.

2) This section should be described in detail. 2.4. NAS score

3) 2.6. Protein Extraction and Western Blot Assay. 2.14. Immunofluorescence. Antibodies should be listed

4) A diagram of the identified mechanism of the cytotoxic effect of iron on liver tissue is needed

5) In the discussion, much attention should be paid to the effects on liver tissue. Discuss possible mechanisms of liver damage with other toxicants and methods of treatment. Antifibrotic Effect of Selenium-Containing Nanoparticles on a Model of TAA-Induced Liver Fibrosis - PubMed

doi:10.24412/2500-2295-2024-3-136-151

6) The conclusion is not valid. Too brief and does not reflect the results obtained

Round 2

Reviewer 1 Report

Comments and Suggestions for Authors

The authors have addressed all the specific points raised in the Reviewer comments. However, two general problems remain.

 1) Heme oxygenase is known to be upregulated by oxidative stress through the redox-sensitive transcription factor Nrf2. In the manuscript, the authors decided to search for alternative regulation mechanism by long non coding RNAs. Although their results are convincing, the widely accepted regulation by Nrf2 must be discussed. 

2) In the Methods section, the extraction of proteins for immunobloting is not described. What was the extraction buffer used? Were the samples reduced and boiled prior to SDS-PAGE?

Reviewer 2 Report

Comments and Suggestions for Authors

The authors have heard all my comments. The article can be accepted in its current form.

Author Response

Dear Reviewer and Editor,

Thank you very much for your valuable comments on our manuscript. Your insightful comments have significantly enhanced the readability and scientific integrity of our work. We have carefully read the manuscript and made revisions based on other comments. Thank you for considering our manuscript for publication in Biology. Once again, we truly appreciate your guidance and support throughout the review process.

Sincerely yours,

Dr. Wenqiang Ma

Key Laboratory of Animal Physiology & Biochemistry

Nanjing Agricultural University

Nanjing, Jiangsu 210095, P. R. China

Email: wq8110@njau.edu.cn

Tel.: 00862584396413